# A Novel Fault-Tolerant Air Traffic Management Methodology Using Autoencoder and P2P Blockchain Consensus Protocol

Seyed Mohammad Hashemi *, Seyed Ali Hashemi, Ruxandra Mihaela Botez  and Georges Ghazi

Laboratory of Applied Research in Active Controls, Avionics, and AeroServoElasticity LARCASE,
ÉTS-École de Technologie Supérieure, Université de Québec, Montréal, QC H3C 1K3, Canada
* Correspondence: seyed-mohammad.hashemi.1@ens.etsmtl.ca

**Abstract:** This paper presents a methodology for designing a highly reliable Air Traffic Management and Control (ATMC) methodology using Neural Networks and Peer-to-Peer (P2P) blockchain. A novel data-driven algorithm was designed for Aircraft Trajectory Prediction (ATP) based on an Autoencoder architecture. The Autoencoder was considered in this study due to its excellent fault-tolerant ability when the input data provided by the GPS is deficient. After conflict detection, P2P blockchain was used for securely decentralized decision-making. A meta-controller composed of this Autoencoder, and P2P blockchain performed the ATMC task very well. A comprehensive database of trajectories constructed using our UAS-S4 Ehécatl was used for algorithms validation. The accuracy of the ATP was evaluated for a variety of data failures, and the high-performance index confirmed the excellent efficiency of the autoencoder. Aircraft were considered in several local encounter scenarios, and their trajectories were securely managed and controlled using our in-house Smart Contract software developed on the Ethereum platform. The Sharding approach improved the P2P blockchain performance in terms of computational complexity and processing time in real-time operations. Therefore, the probability of conflicts among aircraft in a swarm environment was significantly reduced using our new methodology and algorithm.

**Keywords:** air traffic management; trajectory prediction; fault-tolerant Autoencoder; blockchain consensus protocol

## 1. Introduction

Air Traffic Management and Control (ATMC) is one of the most critical issues in aviation transportation [1]. The safety and efficiency of aviation transportation require advanced path planning systems. Various strategies have been designed for path planning to manage airspace [2], traffic flow [3], and capacity [4]. Path planning methods can be categorized into two classes, namely representation and coordinate-related techniques [5]. Representation techniques are based on c-space and include cell decomposition [6], roadmaps [7], potential fields [8], and Voronoi diagrams [9]. The coordinate [10] and non-coordinate techniques are based on various algorithms such as evolutionary models [11], genetic algorithms [12,13], particle swarm optimization [14], and ant colony optimization [15].

Vulnerabilities in an ATMC system, due to faults and failures [16], can result in catastrophic aerial collisions [17]. Safe airspace allocation, therefore, has a key role in ATMC performance [18]. Coordinate-based methodologies have shown very good efficiency in airspace allocation. Linear programming [19], control theory [20,21], meta-heuristic algorithms [22], evolutionary techniques [23], and machine learning [24] models are the most well-known and widely used approaches. The fault-tolerance feature [25] must be considered a fundamental criterion for the design and development of ATMC systems in order to improve safety requirements. Among the above-mentioned methodologies, those based on artificial intelligence (especially deep learning algorithms) have proven their extraordinary ability for fault detection [26], air traffic management [27], control [28], and collision avoidance.

The ATMC system mainly works based on "future trajectories" [29]. By relying on advanced trajectory prediction algorithms, very good decisions can be made, which greatly improves the efficiency of trajectory planning, and thus the performance of the ATMC [30,31]. The first requirement is the design of an accurate Aircraft Trajectory Prediction (ATP) model. Many deterministic and probabilistic approaches have been investigated for the ATP model [24,32–35]. Among these, data-driven algorithms have shown the best performance, particularly when they are provided with a rich dataset [36]. Hence, the first aim of this paper was to design an ATP model based on deep Neural Network architectures. Among them, the Autoencoder algorithm has shown excellent functionality and results for vehicle trajectory prediction. Vehicle (automobile) trajectory prediction with a variational Autoencoder [37], vessel (ship) trajectory prediction using a dual linear Autoencoder [38], and aircraft trajectory prediction using a deep Autoencoder [39] are some examples of very good utility and performance of the Autoencoder. Among the advantages of the Autoencoder, its fault tolerance feature remains its main and outstanding specification [40]. In other words, the Autoencoder can perform accurate trajectory prediction despite deficient (possibly wrong) data obtained from the GPS. This is the reason why we have therefore selected an Autoencoder algorithm for our UAS-S4 trajectory prediction.

In addition to predicting future trajectories, a decision-making algorithm is also needed to manage and control aircraft to avoid collisions. In essence, ATMC is a distributed mission and a central coordinator cannot perform this task properly. By considering a single air traffic controller for a specific flight area, the ATMC system may have difficulties in crowded flight zones. Moreover, reliability issues that arise from failures (dues to miscommunications, adversarial attacks, delayed control commands, or overloaded zones) confirm the advantages of decentralized over centralized approaches [41].

Local broadcast methodology, control zones, trigger, and Peer-to-Peer (P2P) blockchain are among the most widely used fault-tolerant methodologies [42]. Local broadcast uses an efficient and simple algorithm, but it cannot be applied on a grided airspace topology [43]. The control zones approach is fault-tolerant in the case of many agent failures (with high probability), but a global knowledge of aircraft trajectory is required [44]. The trigger strategy can tolerate many agent failures without requiring knowledge of its topology, but in some cases is less efficient than that of control zones [45].

Basically, a vehicle's movement in an environment requires dynamic path planning, as each vehicle's trajectory affects another one. Therefore, a consensus algorithm is needed for analyzing collected trajectory data, and path planning while minimizing the risk of conflict between vehicles. The most successful consensus algorithms have been developed based on deterministic, randomized, leader-free, and leader-based methodologies. Deterministic approaches may not solve the consensus problem due to a single communication failure in an asynchronous environment. Randomized methodologies are not suggested for critical tasks such as air traffic management due to safety issues. The leader-based methodology is more vulnerable than others in the case of cybersecurity attacks. Leader-free methodology outperforms others if the safety issue is the main concern. We need Blockchain as the most reliable infrastructure for implementing a leader-free consensus algorithm.

P2P blockchain is the most secure decentralized coordination methodology that allows the consensus model to make fully collective decisions [46]. As the ATMC system needs a concursus algorithm for reliable airspace allocation, blockchain based on the P2P topology can safely perform this task. Although its high computational complexity is its drawback, by relying on fast processors and desired GPS data collection time, P2P blockchain outperforms other methodologies.

Among the P2P strategies, Linear Consensus Protocol has been widely used for decentralized management and control of network dynamic agents. However, it is sensitive in realistic operation when faults and failures occur on sensors and actuators (hard failures), and when faced with Adversarial or Sybil attacks (soft failures) [47]. Hence, blockchain technology was developed to provide a decentralized, scalable, and secure P2P decision-making system. Ethereum can provide a platform for creating Smart Contracts not only

for financial transactions [48], but also for swarm robots [49], multi-agent UAV moving management [50], and others. Our customized Smart Contract based on the Ethereum framework manages and controls aircraft future trajectories, relying on trajectories predicted by the Autoencoder.

The paper is organized as follows. Section 2 presents the Air Traffic Management problem statement. The methodology for aircraft trajectory prediction using an Autoencoder algorithm with our ATP model and the implementation of the P2P blockchain for the ATMC are presented in Section 3. Section 4 is devoted to the results, including representing the accuracy of the ATP model under data failure, and the performance of the Smart Contract designed for the ATMC. Finally, Section 5 provides a general discussion, conclusions, and recommendations for future work.

## 2. Problem Statement

Assume a flight area composed of airspace-like cubes, with several aircraft flying through their desired trajectories. Figure 1 shows the assumed flight area, constructed by cubes $C_{(x,y,z)}$, in which $u$, $v$, and $w$ are the number of cubes along the $x$, $y$, $z$ axes, respectively.

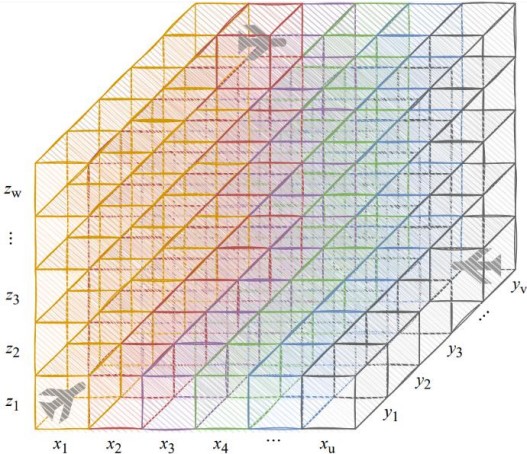

**Figure 1.** Flight area composed of cubic airspaces.

According to Figure 1, there is no limitation to the cube size. The cube size is $x \times y \times z$ in which they are positive integers. The figure is rendered colourful in order to well illustration the airspace which is composed of small cubes. Aircraft might be found in cubes according to their latitude, longitude, and altitude, and they pass through airspaces following their individual trajectories. To avoid aerial collisions, aircraft should never be in the same airspace. Hence, traffic management must allocate safe $C_{(x,y,z)}$ coordinates to aircraft. To improve safety while reducing costs, long-term series of $C_{(x,y,z)}$ allocation is needed in time. Producing such long-term series makes the prediction of future aircraft trajectories an essential requirement for efficient $C_{(x,y,z)}$ allocation. For the prediction of future aircraft trajectory, it is assumed that aircraft fly in their air corridors, as shown in Figure 2. Considering that GPS data (which includes latitude, longitude, altitude, heading, speed, and time of the aircraft) at a given time $T_n$ is available, a model can therefore be used to predict the aircraft's future trajectories for the next $i$ time steps.

By using such predicted trajectories (the first objective), future airspace $C_{(x,y,z)}$ coordinates that minimize the risk of conflict (the second objective) can be allocated to an aircraft by relying on its determined cube-based path. These objectives highlight the need for efficient methodologies for accurate trajectory prediction and reliable airspace allocation.

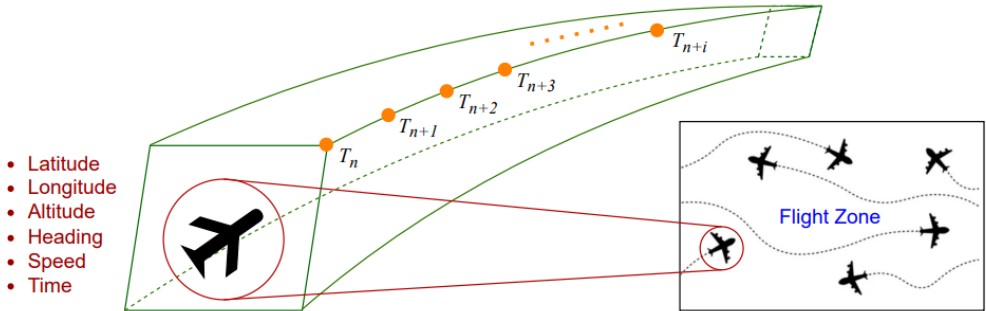

**Figure 2.** Aircraft Trajectory Prediction for Air Traffic Management.

## 3. Methodologies

This section presents the methodologies for realizing accurate trajectory prediction and traffic management. To satisfy the first objective, a data-driven model based on a Neural Network was designed for trajectory prediction. Then, to meet the second objective, a blockchain-based protocol was designed for airspace allocation to ensure safe and efficient traffic management.

### 3.1. Trajectory Prediction

A temporal latent Autoencoder [51] was used as the benchmark algorithm to develop the Aircraft Trajectory Prediction (ATP) model. This Autoencoder is an unsupervised protocol that not only removes noise from the GPS data but also reconstructs compressed data. Following strong dependencies among training data, it can also perform good-quality predictions even when data is deficient. These outstanding features make this Autoencoder an essential tool in fault-tolerant prediction strategies. Figure 3 shows the Autoencoder architecture that has been considered for the ATP model.

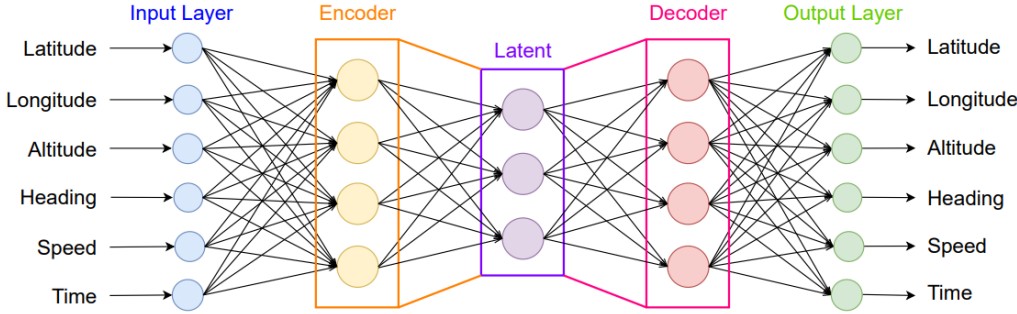

**Figure 3.** The Autoencoder architecture for fault-tolerant aircraft trajectory prediction [51].

As shown in Figure 3, the data provided by the GPS (i.e., latitude, longitude, altitude, heading, speed, and time) is fed to the input layer, which then fully propagates it through the Encoder layer that reduces the dimensions of the input data. The Latent layer in the middle of the architecture produces the "Code", which is fully passed to the Decoder layer. The Decoder is the mirror of the Encoder; its role is to enlarge the Code dimensions, and thereby prepare the full-dimension data for the Output layer. The robustness of the Autoencoder varies with the number of neurons in the Latent layer. The Autoencoder is trained to predict future trajectories even when some elements in the input vector are not well measured. For instance, in Figure 3, by considering 3 neurons in the Latent layer, in the worst case, the Autoencoder will still be able to predict trajectories in the case where there are 3 missing inputs.

The principles of how the Autoencoder works are described next. Let us consider two Euclidean spaces, decoded GPS data $A$, and encoded GPS data $B$. Using decoder $D$, and encoder $E$ transfer functions, $D : B \rightarrow A$ and $E : A \rightarrow B$, where both functions

are multilayer perceptrons [52]. For encoded and decoded data samples, the following statement holds:

$$\text{For any } a \in A \text{ and } b \in B \quad \rightarrow b = E(a) \text{ and } á = D(b) \tag{1}$$

where $a$ is the encoded and $á$ is the decoded GPS data. Training this Autoencoder requires a critical function to evaluate the quality of the training. This task is performed using a predetermined probability distribution $\Omega_{ref}$ over the encoder transfer function $E$, and a quality function $Q = A \times A \rightarrow [0, \infty]$ associated with trajectory reconstruction, such that $d(a, á)$ measures the difference between $a$ and $á$. Therefore, the loss function that must be minimized by solving this optimization problem using the gradient technique can be mathematically represented as follows [52]:

$$L = \hat{E}_{a \sim \Omega_{ref}}[d(a, D(E(a)))] \tag{2}$$

Next, by considering the reference distribution for $\{a_1, \ldots, a_p\}$ such as [52]:

$$\Omega_{ref} = \frac{1}{p} \sum_{h=1}^{p} \delta_{a_h} \tag{3}$$

where $\delta$ is the Dirac measure.

By considering L2 loss: $d(a, á) = \|a - á\|_2^2$ as the Autoencoder quality function (norm 2), the optimal Autoencoder would be formulated as the following least-square optimization problem [52]:

$$\min L \text{ , where } L = \frac{1}{p} \sum_{h=1}^{p} \|a_h - D(E(a_h))\|_2^2 \tag{4}$$

The hyperparameters of the trained Autoencoder for ATP were fine-tuned by minimizing the optimization criteria given in Equation (4). The trajectories predicted by the Autoencoder can then be used for airspace allocation and traffic management.

### 3.2. Airspace Allocation

By using predicted trajectories, a receding-horizon lattice planner was developed for dynamic airspace allocation [53]. A Consensus Protocol must confirm that an aircraft will not be in a conflicting (possible collision) zone in any future steps [36]. To guarantee separation between aircraft, the Peer-to-Peer (P2P) topology [54] shown in Figure 4 was used as a decentralized Consensus Protocol based on blockchain technology.

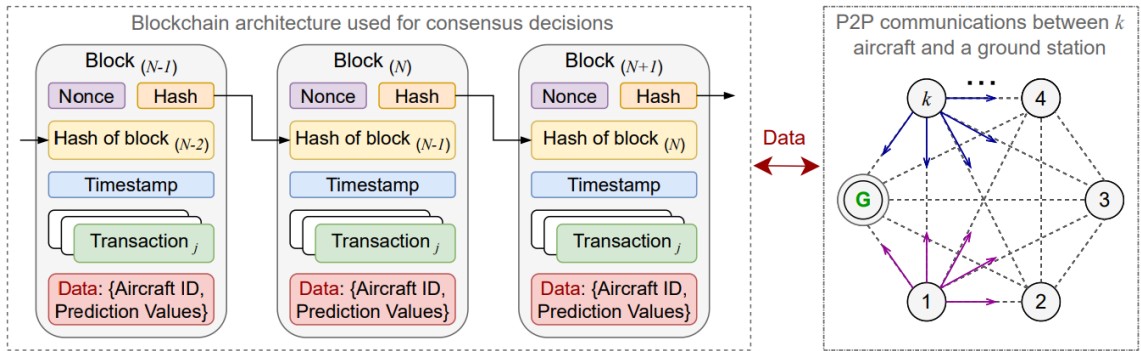

**Figure 4.** P2P blockchain architecture that supports smart contracts for the consensus protocol.

Regardless of blockchain methodology, utilizing the P2P topology provides robust communication in case of data loss. Blockchain is an outstanding methodology for coming up with wrong data. Hence, using the blockchain methodology, which is developed based

on P2P topology, not only provides robustness in case of data loss but also in case of wrong data (predicted by the Autoencoder) due to adversarial attacks. The Ethereum-based Smart Contract [55] executes Block$_{(N+1)}$ that allows aircraft to follow their trajectories if there is no conflict between the trajectories predicted by their corresponding Autoencoders.

As illustrated on the right-hand side of Figure 4, a ground station and $k$ aircraft are assumed, which are fully connected and communicate with each other, and a P2P topology is therefore constructed. Each aircraft can be considered an "agent", and is equipped with an Autoencoder that can be used to predict its future trajectories. Each aircraft submits its predicted trajectory (using its own Autoencoder) to the blockchain by itself. In the order word, each aircraft is an agent that creates a node on the P2P blockchain topology. All aircraft should confirm the future trajectories of the others relying on the previous executed block in the blockchain that contains all aircraft future trajectories in the last timestamp. The ATP is performed decentralized, and its values are shared among agents through the P2P topology. Blockchain methodology keeps trajectory data private and only allows identified aircraft to read and write the data on the chain. Each aircraft is considered a node and must individually act as an agent, share its data with others nodes, and write it on the blockchain. Block$_{(N+1)}$ is executed when the Smart Contract confirms that there is no conflict between aircraft relying on the Autoencoders and Block$_{(N)}$. Secured data mainly includes the aircraft identifier (ID) $[1 - k]$, and its predicted trajectories which are stored in Block$_{(N+1)}$. This process is performed on-chain, relying on the previous block hash ("data encryptor"), the added number to the hash ("nonce"), exact time execution ("timestamp"), and $j$ number of agreements for trajectory predictions ("transactions"). Trajectory Data are logged once an aircraft enters a district, and lasts as long as it exists in the corresponding districts.

Increasing the number of aircraft in a flight zone is a real challenge. Sharing data among many aircraft increases both the computational complexity and the error rate for a Smart Contract. Sharding means we consider aircraft in some classes regarding their close locations. The Smart Contract cannot execute new blocks quickly, and it may fail in the Block execution for unnecessary agreement. The concept of Sharding (scaling of the blockchain network) allows the Smart Contract to categorize aircraft into multiple districts (sub-flight zones) in terms of their locations. The ground station communicates with the districts through sharding and each of which oversees the path planning for $l_m$ aircraft, as shown in Figure 5.

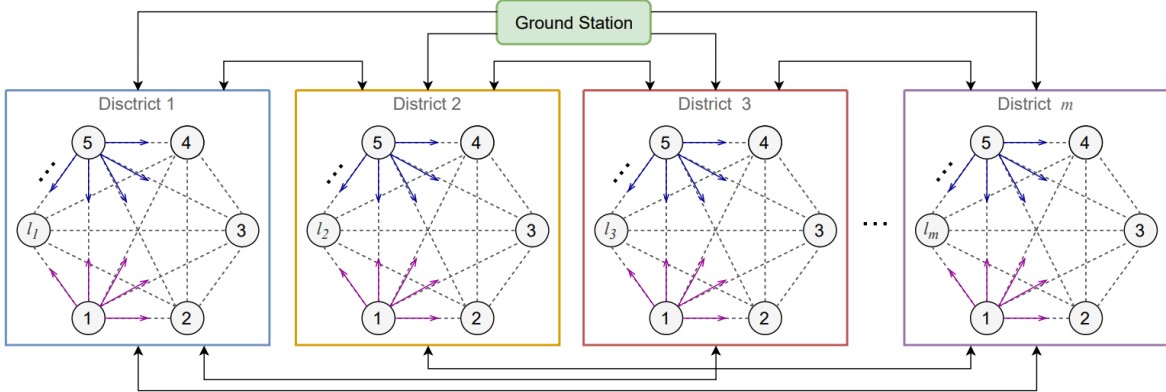

**Figure 5.** Sharding a flight zone into several districts for edge computing in each sub-flight zone.

Figure 5 illustrates the P2P communication topology between aircraft. The flight zone is sharded by considering $m$ districts. Each district creates a sub-flight zone that contains $l_m$ agents (aircraft), where $m$ indicates the number of sub-flight zones (districts). In other words, $k$ aircraft in a flight zone are sharded into $m$ sub-flight zones according to their locations. Worth mentioning that nodes 1 to 5 in each district are unique. For instance, node number 1 in district number 1 is not the same as node number 1 in district number 2, and each one represents the node corresponding to a particular aircraft.

The ground station does not communicate directly with the aircraft, but instead calls Smart Contract for the required data that are obtained from each individual district. The sharding methodology can reduce the computational complexity of the Smart Contract while reducing the error rate due to failed agreements associated with unnecessary encounter situations. The following section explains and discusses these improvements from a numerical perspective.

## 4. Results and Discussion

To test and validate the methodology presented in this study, aircraft trajectories were generated using our UAS-S4 flight dynamics model [56,57], and its corresponding controller [58]. This aircraft is pictured in Figure 6, and its specifications are listed in Table 1. A number of 1820 flight trajectories containing 218,400 samples (i.e., [*latitude*, *longitude*, *altitude*, *heading*, *speed*, *time*]$_{6\times1}$) were generated and used for the training, testing, and validation of our newly designed Autoencoder.

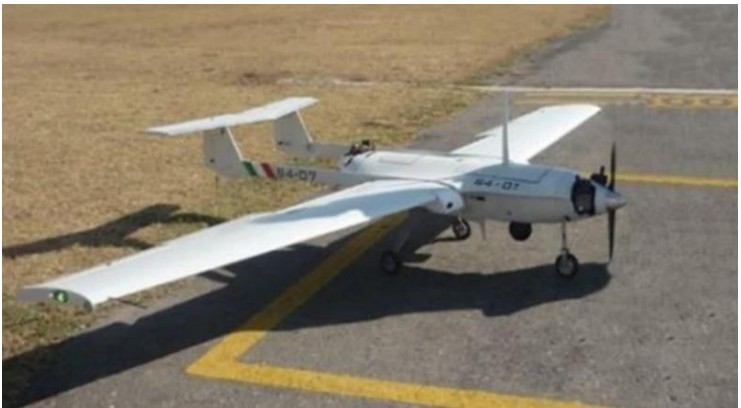

**Figure 6.** Hydra Technologies UAS-S4 Ehecatl.

**Table 1.** The UAS-S4 geometrical and flight data specifications.

| Specification | Value |
| --- | --- |
| Wingspan | 4.2 m |
| Wing area | 2.3 m$^2$ |
| Total length | 2.5 m |
| Mean aerodynamic chord | 0.57 m |
| Empty weight | 50 kg |
| Maximum take-off weight | 80 kg |
| Loitering airspeed | 35 knots |
| Maximum speed | 135 knots |
| Service ceiling | 15,000 ft |
| Operational range | 120 km |

Weighting vectors were initialized, and then a batch normalization was performed by recentering and rescaling the input data with the aim to obtain a fast and stable trajectory prediction [59]. Regularization was also performed, as a means to reduce the magnitudes of regressed trajectories [60]. The hyperparameters were tuned [61]; with a Code size of $3 \times 1$ (vector), the system could be tolerant in the case of faults or failures of three out of all six elements of the input vector [*latitude*, *longitude*, *altitude*, *heading*, *speed*, *time*]$_{6\times1}$.

For the encoder, 4 and 5 numbers of nodes were considered, and the decoder architectures were the same as those of the encoder. The Mean Absolute Error (MAE) was determined as the loss function. Table 2 presents the prediction accuracy, and precision of the Autoencoder, which is compared with the Long Short-Term Memory (LSTM) [36] under four different data failure combinations.

**Table 2.** The Fault-tolerant Autoencoder and LSTM trajectory prediction accuracy and precision comparison.

| Data Failures | Prediction Accuracy % | | Precision | |
|---|---|---|---|---|
| | LSTM | Autoencoder | LSTM | Autoencoder |
| No failures | 99.5 | 98.7 | 0.01 | 0.01 |
| Latitude | 91.6 | **98.1** | 0.04 | **0.03** |
| Latitude and heading | 83.2 | **95.3** | 0.06 | **0.04** |
| Latitude, altitude, and speed | 58.7 | **91.2** | 0.10 | **0.07** |

According to Table 2, the LSTM shows slightly better performance than the Autoencoder only in the absence of failure. However, when these methods use deficient data (e.g., latitude, altitude, and speed failure), the Autoencoder shows very good prediction accuracy (91.2%), while the LSTM shows a much worse prediction accuracy (58.7%). These results confirm the excellent fault-tolerant ability of the Autoencoder. Worth mentioning that the training trajectories data were generated using the simulator while random disturbances such as wind shear, wind gusts, and turbulences were considered in the simulation process. In accordance with Table 2, 98.7% prediction accuracy in the test phase confirms the generalizability of the Autoencoder for trajectory prediction even in case of unknown disturbances dues to climate changes.

Moreover, the Autoencoder performance is dependent on its architecture (number of neurons in the Encoder, Latent, and Decoder) as shown in Table 3. For this purpose, we considered different feasible numbers of neurons for Encoder, Latent, and Decoder to determine which number gives the best performance.

**Table 3.** The Autoencoder performance considering different number of neurons in the architectures.

| Data Failures | Number of Neurons in the Encoder and Decoder | Number of Neurons in the Latent | Prediction Accuracy % | Error Rate % |
|---|---|---|---|---|
| Latitude | 5 | 4 | 97.8 | 4.2 |
| | | 3 | 97.6 | 4.1 |
| | 4 | 3 | 98.1 | 3.3 |
| | | 2 | 43.3 | 17 |
| Latitude and Heading | 5 | 4 | 94.9 | 5.2 |
| | | 3 | 94.6 | 6.3 |
| | 4 | 3 | 95.3 | 4.4 |
| | | 2 | 39.2 | 16.6 |
| Latitude, Altitude, and Speed | 5 | 4 | 90.8 | 8.5 |
| | | 3 | 90.3 | 9.2 |
| | 4 | 3 | 91.2 | 7.1 |
| | | 2 | 32.6 | 19.8 |

Table 3 shows that using 2 neurons in the Latent layer when 3 elements (i.e., latitude, altitude, and speed) in the input vector were lost (had failures) resulted in the Code generating failure, and dramatic prediction accuracy reduction (shown in red). It is worth noting that the use of 4 neurons in the Encoder layer can provide better data for coding than using 5 neurons. Prediction Accuracy (shown in green) confirms that the best Autoencoder architecture is composed of 3 neurons in the Latent layer, and 4 neurons in the Decoder and Encoder layers.

Relying on the given fault-tolerant Autoencoder for trajectory prediction, 120 UAS-S4s were arranged in a flight zone, in which each had its own location, heading, and speed. Their future trajectories were managed and controlled using the Smart Contract. A private chain combining the "consensus layer" with the "contract layer" was designed for ATMC decision-making, based on proof of time and Smart Contract approaches, respectively. We considered the Linear Consensus Protocol (LCP) as the baseline and compared its

performance to the performance of our Smart Contract-based Consensus Protocol (SCCP). Both methodologies were analyzed in cases of "soft" failures (Sybil Attacks and Adversarial Attack). In order to improve Smart Contract performance to avoid unnecessary encounter situations, the Sharding methodology was used to design several sub-flight zones. Table 4 shows the error rate while different Consensus Protocols were managing the UAS-S4′s trajectories and different attacks were imposed on the aircraft.

**Table 4.** ATMC error rate when consensus protocols control and manage traffic.

| Methodology | Number of Sub-Flight Zones | Attacks Imposed? | Error Rate % |
|---|---|---|---|
| Linear Consensus Protocol (LCP) | 1 | No | 5.3 |
|  |  | Yes | 83.2 |
| Smart Contract-based Consensus Protocol (SCCP) | 1 | No | 8.7 |
|  |  | Yes | 18.6 |
| Sharding-SCCP | 4 | No | 7.7 |
|  |  | Yes | **15.4** |

According to the Error Rate shown in Table 4, when soft attacks are not imposed, the Linear Consensus Protocol (LCP) methodology can perform the ATMC task with the smallest error rate (5.3%). However, the superiority of the Smart Contract-based Consensus Protocol (SCCP) and Sharding-SCCP over the LCP becomes obvious when Sybil Attacks and Adversarial Attacks are considered. Under such conditions, the SCCP showed an 18.6% error rate, while the LCP almost collapsed (83.2% error rate). Sharding improved the SCCP performance (down to a 15.4% error rate). The following section elaborates on and summarizes these results.

For the reliability evaluation, we considered the error rate as the performance index. The error rates due to block execution failures were measured, while various consensus protocols were approaching adversarial attacks. Worth mentioning that the randomized consensus methodology is not utilized in the comparison study due to its inherent safety weakness for performing the ATM critical task. The block execution error rates were 2.24%, 2.09%, and 0.87% when deterministic, leader-based, and leader-free consensus methodologies were in charge, respectively. These error rate values confirm the leader-free blockchain methodology is more reliable than other consensus protocols while approaching adversarial attacks.

The probability of UAS conflicts after airspace allocation might be analyzed as a critical dynamic performance. A total of 100 UAS-S4s were operated at $z = 3.75$ km (altitude), $x = 4$ to 7.2 km (latitude), and $y = 8$ to 9.2 km (longitude), by utilizing 16 shards to ensure safe performance. Figure 7 illustrates the probability of conflicts among three UAS-S4s in a specific sub-flight zone under two protocols, the LCP (a) and blockchain (b).

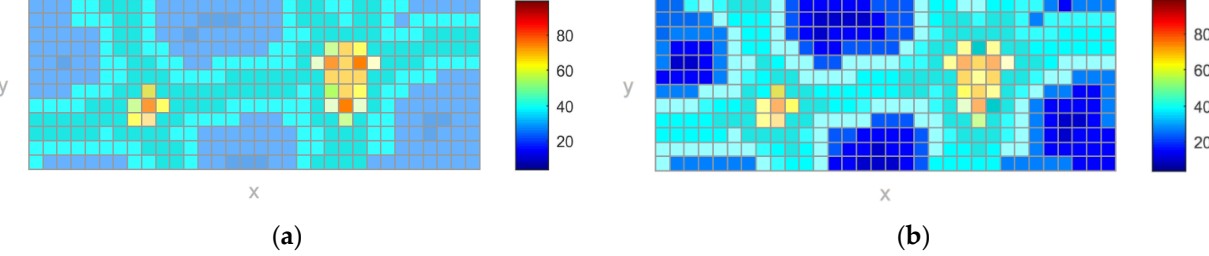

(**a**)　　　　　　　　　　　　　　　　　　　　　　　(**b**)

**Figure 7.** Probability of UASs conflicts using Linear Consensus Protocol LCP (**a**) and blockchain (**b**).

Figure 7 is extracted from the 10th shard, where 3 UASs have conflicts within 3.84 km$^2$. A comparison between plot (a) for the LCP and plot (b) with blockchain confirms that the blockchain performance was better than that of the LCP in terms of the probabilities of conflicts and safety. The average probability of conflicts was 33.1% for the LCP and 26.7%

for the blockchain. In addition, the LCP does not give safe zones, while the blockchain approach offers a substantial number of such zones.

## 5. Conclusions

A reliable Air Traffic Management and Control (ATMC) was designed based on an Autoencoder, and a Smart Contract on the Ethereum blockchain. The Autoencoder was designed to predict future flight trajectories despite partial failures in the GPS data. It has shown excellent prediction accuracy when tested with deficient data obtained from the GPS. The impact of the number of neurons used in the Autoencoder's hidden layers on its performance was investigated. The number of neurons in the Latent layer determined the threshold robustness for the Autoencoder when it is faced with deficient data. Relying on the predicted trajectories using this fault-tolerant Autoencoder, a Consensus Protocol was designed for Air Traffic Management and Control (ATMC). A Smart Contract based on Ethereum blockchain was designed to perform the ATMC task. The Smart Contract Consensus Protocol (SCCP), and a Sharding SCCP showed very good performance under Sybil Attacks and Adversarial Attacks. The designed ATMC system composed of the Autoencoder, and the Smart Contract Consensus Protocol showed very good robustness in cases of data failures or attacks.

**Author Contributions:** Conceptualization, S.M.H. and S.A.H.; methodology, S.M.H. and S.A.H.; software, S.M.H. and S.A.H.; validation, S.M.H., S.A.H., R.M.B. and G.G.; formal analysis, S.M.H., S.A.H., R.M.B. and G.G.; investigation, S.M.H. and S.A.H.; resources, R.M.B. and G.G.; data curation, S.M.H.; writing—original draft preparation, S.M.H.; writing—review and editing, R.M.B. and G.G.; visualization, S.M.H.; supervision, R.M.B. and G.G.; project administration, R.M.B.; funding acquisition, R.M.B. and G.G. All authors have read and agreed to the published version of the manuscript.

**Funding:** This research was funded by NSERC within the Canada Research Chairs program, which made possible the realization of this research and the publication of this paper. Ruxandra Botez is the Canada Research Chair Tier 1 Holder in Aircraft Modeling and Simulation New Technologies.

**Data Availability Statement:** Not applicable.

**Acknowledgments:** Special thanks are due to the Natural Sciences and Engineering Research Council of Canada (NSERC) for the Canada Research Chair Tier 1 in Aircraft Modeling and Simulation Technologies funds. We would also like to thank Odette Lacasse and Oscar Carranza for their support at ETS, as well as Hydra Technologies' team members Carlos Ruiz, Eduardo Yakin, and Alvaro Gutierrez Prado in Mexico. Finally, we wish to express our appreciation to the Canada Foundation for Innovation CFI, the Ministère de l'Économie et de l'Innovation and Hydra Technologies for their support of the acquisition of the UAS-S4 at the LARCASE.

**Conflicts of Interest:** The authors declare no conflict of interest. The funders had no role in the design of the study; in the collection, analyses, or interpretation of data; in the writing of the manuscript, or in the decision to publish the results.

## Nomenclature

| | |
|---|---|
| $A$ | Decoded GPS data |
| $B$ | Encoded GPS data |
| $E$ | Encoder transfer function |
| $D$ | Decoder transfer function |
| $i$ | Number of steps for future trajectory prediction |
| $j$ | Number of executed agreements |
| $k$ | Number of aircraft in a flight zone before sharding |
| $l$ | Number of aircraft in a flight zone after sharding |
| $L$ | Autoencoder loss function |
| $m$ | Number of districts that provide sub-flight zones after sharding |
| $N$ | Number associated with a block in the chain |
| $Q$ | Autoencoder quality function |
| $T_n$ | Time during which the aircraft is in step $n$ |

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
