# Peer review of "A Novel Fault-Tolerant Air Traffic Management Methodology Using Autoencoder and P2P Blockchain Consensus Protocol"

_aerospace, doi:10.3390/aerospace10040357_

Round 1
Reviewer 1 Report
Brief summary
The authors aim to achieve two main goals here:
- firstly, they want to design an Air Traffic Prediction model based on deep Neural Network architectures, and more precisely on the use of auto encoder algorithm due to its good performance in fault-tolerant environments (which might be the case in Air Traffic Management systems);
- secondly, they want to propose a decision-making algorithm based on Blockchains to distributively resolve potential conflicts in aircraft trajectories.
General concept comments
The paper is engaging, figures are aesthetically pleasant, authors provide a strong case for their study and I agree with the necessity to find alternative protocols to alleviate Air Traffic Controllers from decision-making.
However, I find the paper rather vague on the technicalities. While the justifications for the work are limpid, the use of blockchain or blockchain-inspired consensus algorithm (?) leaves me wondering why do authors need a blockchain (of which the architecture is roughly described in Figure 4). The smart contract is only evoked - it would have greatly improve my understanding on how blockchain is actually supporting this work, and eliminate my believe that this work would have been possible just fine with a more traditional consensus algorithm. Maybe authors should emphasize on the « reliability » property they claim to achieve by defining its meaning in the context of ATMC.
Overall, the absence of data description is too important to overlook, especially since we are considering a data collection and sharing platform, data on which some may take decisions on. The lack of information makes it impossible to reproduce the work and check the validity of the results.
Specific comments
The introduction is clear, comprehensive and identify adequately the gap identified by authors (which falls in -in my opinion - in the category of optimization works).
Lines 83—87
What makes Blockchain so suitable to ATMC systems? I don’t understand how [46] justifies the claim « P2P Blockchain is the most secure decentralized coordination methodology that allows the ATMC system to make 84 fully collective decisions46. »
Isn’t there any consideration or hypothesis to make regarding data loss and its compliance with blockchain security requirements?
Lines 91-97 and section 3.2 and table 4:
Makes me feel that the blockchain technology is kind of reduced to its consensus algorithm. There are little or vague explanations on the data aspect of blockchains which make the interpretation of Table 4 complicated or impossible:
- What types of data is logged ? Size of data? Cost to log it into a blockchain? For how long?
- What type of blockchain this is? Are the data public? Who can write data into the blockchain?
- How does one submit data to the blockchain?
- What types of attacks are your applying on the consensus protocols…
The Problem Statement section reviews precisely the challenges and describe the context of the work.
Figure 1:
The size of Figure 1 could be reduced. Why 5x5x5 cubes? Is there a meaning for the colors (uncertainty, …)?
The Methodologies section explicit the use of autoencoder. I am though left wondering about training data: with climate change, will a AI-based trajectory planning still be relevant (considering that meteorological events may disturb trajectory patterns)?
Figure 3: what is the input (lat, long,…) but most importantly when are they taken? Moreover, who computes the whole trajectories? And again, who submits them to the blockchain, validates, etc…
I believe there is a font issue at lines 169-173.
Figure 5 does not help with the understanding of sharding. Are nodes 1 to 5 the same in all districts? What do the arrows refer to?
Section 4 (results and discussion) shows promising results, but for all the aforementioned reasons, I find it difficult to trust and interpret.
Lines 271-290 are the ones that try to explain the use of blockchain but it comes short and late in the paper (especially with a title that contains the word blockchain).
Maybe Figure 6 and table 1 could be side by side (that would save space for more details).
Author Response
Journal: Aerospace (MDPI)
Ref manuscript ID Aerospace-2284040
Manuscript title “A Novel Fault-Tolerant Air Traffic Management Methodology using Autoencoder and P2P Blockchain Consensus Protocol”.
Dear Editor and Reviewers, we (the authors) would like to thank you very much for your constructive comments, that helped us to improve the writing of this paper, that we are submitting for its publication in the ‘Aerospace’ journal. Please find our answers (A) to reviewers’ questions (Q) below:
Answers of the authors to reviewer #1 questions
We would like to thank very much to the reviewer #1 for his or her revisions of this paper, that were helpful in the writing of this paper. Please find our answers (A) to your questions (Q) below:
Reviewer #1:
Q.1 Why do authors need a blockchain (of which the architecture is roughly described in Figure 4. It would have greatly improve my understanding on how blockchain is actually supporting this work, and eliminate my believe that this work would have been possible just fine with a more traditional consensus algorithm?
A.1 Basically, vehicles' movement in an environment requires dynamic path planning, as each vehicle's trajectory affects another one. Therefore, a consensus algorithm is needed for analysing collected trajectory data, and path planning while minimizing the risk of conflict between vehicles. The most successful consensus algorithms have been developed based on deterministic, randomized, leader-free, and leader-based methodologies. Deterministic approaches may not solve the consensus problem due to a single communication failure in an asynchronous environment. Randomized methodologies are not suggested for critical tasks such as air traffic management due to safety issues. The leader-based methodology is more vulnerable than others in the case of cybersecurity attacks. Leader-free methodology outperforms others if the safety issue is the main concern. We need Blockchain as the most reliable infrastructure for implementing a leader-free consensus algorithm (Lines 84-95 of the revised version).
Q.2 Lines 83-87 (reviewed version):
- Q.2.1 What makes Blockchain so suitable to ATMC systems? I don’t understand how [46] justifies the claim « P2P Blockchain is the most secure decentralized coordination methodology that allows the ATMC system to make fully collective decisions.
- A.2.1 That sentence may be confusing so it is rearranged as the following: P2P Blockchain is the most secure decentralized coordination methodology that allows the consensus model to make fully collective decisions46. As the ATMC system needs a consensus algorithm for reliable airspace allocation, Blockchain based on the P2P topology can safely perform this task (Lines 96-99 of the revised version).
- Q.2.2 Isn’t there any consideration or hypothesis to make regarding data loss and its compliance with blockchain security requirements?
- A.2.2 Regardless of blockchain methodology, utilizing the P2P topology provides robust communication in case of data loss. Blockchain is an outstanding methodology for coming up with wrong data. Hence, using the blockchain methodology, which is developed based on P2P topology, not only provides robustness in case of data loss, but also in case of wrong data (predicted by the Autoencoder) due to adversarial attacks (Lines 206-210 of the revised version).
Q.3 Lines 91-97 (reviewed version) and section 3.2 and table 4
- Q.3.1 What types of data is logged? Size of data?
- A.3.1 It is the GPS data as [latitude, longitude, altitude, heading, speed, time] 6x1. (Lines 263-266 of the revised version).
- Q.3.2 For how long?
- A.3.2 Data are logged once an aircraft enters a district, and lasts as long as it exists in the corresponding districts (Lines 234-235 of the revised version).
- Q.3.3 What type of blockchain this is?
- A.3.3 The Ethereum blockchain was used as the consensus protocol (Line 210 of the revised version)
- Q.3.4 Are the data public?
- A.3.4 It’s It is the strength of the blockchain methodology that keeps data private. Blockchain only allows identified aircraft to read and write the data on the chain (Lines 217-218 of the revised version).
- Q.3.5 Who can write data into the blockchain?
- A.3.5 Each aircraft is considered a node and must individually act as an agent, and share its data with other nodes and write it on the blockchain (Lines 226-228 of the revised version).
- 3.6 What types of attacks are your applying on the consensus protocols?
- 3.6 Sybil attacks are imposed on the consensus protocol and adversarial attacks are imposed on the autoencoder (Line 105 of the revised version).
Q.4 Figure 1:
- Q.4.1 The size of Figure 1 could be reduced.
- A.4.1 The size of the figure is shrunk (Line 126 of the revised version).
- Q.4.2 Why 5x5x5 cubes?
- A.4.2 There is no limitation for the cube size. According to Figure 1, the cube size is in which they are positive integers (Lines 128-129 of the revised version).
- Q.4.3 Is there a meaning for the colors (uncertainty, …)?
- A.4.3 The figure is rendered colorful in order to well illustration the airspace which is composed of small cubes (Lines 129-130 of the revised version).
Q.5 The Methodologies section explicit the use of autoencoder. I am though left wondering about training data: with climate change, will an AI-based trajectory planning still be relevant (considering that meteorological events may disturb trajectory patterns)?
A.5. Training trajectory data was generated using the simulator while random disturbances such as wind shear, wind gusts, and turbulences were considered in the simulation process. In accordance with Table 2, 98.7% prediction accuracy in the test phase confirms the generalizability of the Autoencoder for trajectory prediction even in case of unknown disturbances dues to climate changes (Lines 228-293 of the revised version).
Q.6 Figure 3:
- Q.6.1 What is the input (lat, long, …) but most importantly when are they taken?
- A.6.1 These are the GPS data that represent the aircraft locations. The latitude represents the distance north or south of the equator. The longitude represents the distance east or west of the prime meridian. Altitude is the distance above sea level.
- Q.6.2 Moreover, who computes the whole trajectories?
- A.6.2 The trained Autoencoder is in charge of computing and predicting the whole trajectory of the aircraft at each timestamp (Line 177 of the revised version).
- Q.6.3 And again, who submits them to the blockchain, validates, etc…?
- A.6.3 Each aircraft submits its predicted trajectory (using its own Autoencoder) to the blockchain by itself. In the order word, each aircraft is an agent that creates a node on the P2P blockchain topology. All aircraft should confirm the future trajectories of the others relying on the previous executed block in the blockchain that contains all aircraft future trajectories in the last timestamp (Line 219-224 of the revised version).
Q.7 I believe there is a font issue at lines 169-173 (reviewed version).
A.7 The font is right. Maybe your MS-Word software is not updated.
Q.8 Figure 5 does not help with the understanding of sharding. Are nodes 1 to 5 the same in all districts? What do the arrows refer to?
A.8 Sharding means we consider aircraft in some classes regarding their close locations (Line 238-239 of the revised version). Nodes 1 to 5 in each district are unique. For instance, node number 1 in district number 1 is not the same as node number 1 in district number 2, and each one represents the node corresponding to a particular aircraft (Line 251-253 of the revised version).
The flight zone is sharded by considering m districts. Each district creates a sub-flight zone that contains lm agents (various numbers of aircraft), where m indicates the number of sub-flight zones (districts). In other words, k aircraft in a flight zone are sharded into m sub-flight zones according to their locations (Line 247-251 of the revised version).
Q.9 Lines 271-290 (reviewed version) are the ones that try to explain the use of blockchain but it comes short and late in the paper (especially with a title that contains the word blockchain).
A.9 The answer to question Q.1 is now provided in the context to solve this issue (Lines 84-95 of the revised version).
Q.10 Maybe Figure 6 and table 1 could be side by side (that would save space for more details).
A.10 Figure 6 and Table 1 are arranged side by side (Line 284 of the revised version).

Reviewer 2 Report
The authors presented the details of an interesting approach to support Air Traffic Management and Control (ATMC). The solution proposed is sound, and the paper is interesting to the general public. I believe it can be accepted given some points:
Strengths
- Figures are very well-design and simplify the concepts presented;
- The paper is well-written and presents a promising strategy that can tackle current issue and also be extended to future paradigms;
- The use of LSTM and AE is suitable to the problem considered.
Weaknesses
- The title says “A Novel Fault-Tolerant Air Traffic Management Methodology“. ATM is a broad area with several subareas. It seems to be more reasonable to reshape the paper in a way to clearly define the goal without being so broad. The authors emphasize “A novel data-driven algorithm was designed for Aircraft Trajectory Prediction (ATP)“. If this is the main contribution, ATP should be highlighted instead of ATM.
- The authors claim that the proposal is highly reliable. This is not clear to the reader as reliability can be evaluated using different methods.
- The authors should present a more in-depth discussion on how this proposal compares to the existing solutions.
Author Response
Journal: Aerospace (MDPI)
Ref manuscript ID Aerospace-2284040
Manuscript title “A Novel Fault-Tolerant Air Traffic Management Methodology using Autoencoder and P2P Blockchain Consensus Protocol”.
Dear Editor and Reviewers, we (the authors) would like to thank you very much for your constructive comments, that helped us to improve the writing of this paper, that we are submitting for its publication in the ‘Aerospace’ journal. Please find our answers (A) to reviewers’ questions (Q) below:
Answers of the authors to reviewer #2 questions
We would like to thank very much to the reviewer #2 for his or her revisions of this paper, that were helpful in the writing of this paper. Please find our answers (A) to your questions (Q) below:
Reviewer #2:
Figures are very well-design and simplify the concepts presented; The paper is well-written and presents a promising strategy that can tackle current issue and also be extended to future paradigms; The use of LSTM and AE is suitable to the problem considered.
Q.1 The title says “A Novel Fault-Tolerant Air Traffic Management Methodology “. ATM is a broad area with several subareas. It seems to be more reasonable to reshape the paper in a way to clearly define the goal without being so broad. The authors emphasize “A novel data-driven algorithm was designed for Aircraft Trajectory Prediction (ATP)”. If this is the main contribution, ATP should be highlighted instead of ATM.
A.1 In this paper, we had two main objectives. Firstly, we designed an Aircraft Trajectory Prediction model using a deep Neural Network (Autoencoder) algorithm due to its excellent performance in fault-tolerant environments (which was the case in Air Traffic Management systems). Secondly, a decision-making algorithm based on Blockchains was designed to distributively resolve potential conflicts and manage aircraft trajectories. That is why the "Air traffic management" phrase is addressed instead of "aircraft trajectory prediction" in the title (Lines 151-155 of the revised version).
Q.2 The authors claim that the proposal is highly reliable. This is not clear to the reader as reliability can be evaluated using different methods. The authors should present a more in-depth discussion on how this proposal compares to the existing solutions.
A.2 The most successful consensus algorithms have been developed based on deterministic, randomized, leader-free, and leader-based methodologies. Deterministic approaches may not solve the consensus problem due to a single communication failure in an asynchronous environment. Randomized methodologies are not suggested for critical tasks such as air traffic management due to safety issues. The leader-based methodology is more vulnerable than others in case of cybersecurity attacks. Leader-free methodology outperforms others if the safety issue is the main concern (Lines 87-95 of the revised version).
For the reliability evaluation, we considered the error rate as the performance index. The error rates due to block execution failures were measured, while various consensus protocols were approaching adversarial attacks. Worth mentioning that the randomized consensus methodology is not utilized in the comparison study due to its inherent safety weakness for performing the ATM critical task. The block execution error rates were 2.24%, 2.09%, and 0.87% when deterministic, leader-based, and leader-free consensus methodologies were in charge, respectively. These error rate values confirm the leader-free blockchain methodology is more reliable than other consensus protocols while approaching adversarial attacks (Lines 335-343 of the revised version).
Q.3 Extensive editing of English language and style required
A.3 The paper is edited by a native editor once again.

Round 2
Reviewer 1 Report
Thank you for taking the time to address my comments and those of the other reviewers. These clarifications suit me well.
Two last comments for you to maybe consider.
- Q.3.3 What type of blockchain this is?
- A.3.3 The Ethereum blockchain was used as the consensus protocol (Line 210 of the revised version)
- Q.3.5 Who can write data into the blockchain?
- A.3.5 Each aircraft is considered a node and must individually act as an agent, and share its data with other nodes and write it on the blockchain (Lines 226-228 of the revised version).
1. How do you handle aircraft registration?
2. Will you open source you smart contracts?
Reviewer 2 Report
Thank you for working on this new version. I recommend this manuscript be published as is.